# Multimodal Patient Representation Learning with Missing Modalities and Labels

**Zhenbang Wu**[1,2,3]*, **Anant Dadu**[2,3], **Nicholas Tustison**[4], **Brian Avants**[5],
**Mike Nalls**[2,3], **Jimeng Sun**[1], **Faraz Faghri**[2,3]
[1]University of Illinois Urbana-Champaign, [2]National Institutes of Health
[3]Data Tecnica International, [4]University of Virginia, [5]University of Pennsylvania
zw12@illinois.edu

## Abstract

Multimodal patient representation learning aims to integrate information from multiple modalities and generate comprehensive patient representations for subsequent clinical predictive tasks. However, many existing approaches either presuppose the availability of all modalities and labels for each patient or only deal with missing modalities. In reality, patient data often comes with both missing modalities and labels for various reasons (i.e., **the missing modality and label issue**). Moreover, multimodal models might over-rely on certain modalities, causing suboptimal performance when these modalities are absent (i.e., **the modality collapse issue**). To address these issues, we introduce MUSE: a mutual-consistent graph contrastive learning method. MUSE uses a flexible bipartite graph to represent the patient-modality relationship, which can adapt to various missing modality patterns. To tackle the modality collapse issue, MUSE learns to focus on modality-general and label-decisive features via a mutual-consistent contrastive learning loss. Notably, the unsupervised component of the contrastive objective only requires self-supervision signals, thereby broadening the training scope to incorporate patients with missing labels. We evaluate MUSE on three publicly available datasets: MIMIC-IV, eICU, and ADNI. Results show that MUSE outperforms all baselines, and MUSE+ further elevates the absolute improvement to ~4% by extending the training scope to patients with absent labels.

## 1 Introduction

Patient data spans a wide range of modalities, including images (Johnson et al., 2019; Jack et al., 2008), text (Johnson et al., 2023), physiological signals (Jing et al., 2018; Kemp et al., 2000), and demographics (Pollard et al., 2018; Kahn, 1994). Physicians often jointly consider information from several modalities to make informed decisions about a patient's diagnosis, treatment, and ongoing care. For instance, tracking the progression of Alzheimer's Disease requires diverse data such as clinical scores, genetic profiles, neuroimaging scans, and biomarker readings (Khachaturian, 1985; Veitch et al., 2019). Multimodal patient representation learning aims to integrate information from these diverse modalities and generate comprehensive patient representation for downstream clinical predictive tasks like disease progression, mortality prediction, and readmission prediction (Kline et al., 2022).

**Conventional multimodal learning** typically presumes that all modalities are available for every patient (Ramachandram & Taylor, 2017; Xu et al., 2023). However, in reality, patient data might be missing certain modalities. For example, some Alzheimer's patients may not have genetic or neuroimaging data due to accessibility issues or cost concerns (Schott & Bartlett, 2012). Further, fragmented clinical records are common among patients switching healthcare providers. In such cases, conventional multimodal approaches often disregard these incomplete data, which largely reduces the number of training samples and narrows their application to only patients with complete modalities. To address this, **multimodal learning with missing modalities** (Ma et al., 2021; Chen & Zhang, 2020; Zhang et al., 2022) is proposed. Among them, modality imputation (Tran et al., 2017;

---

*Work done during an internship at National Institutes of Health and Data Tecnica International.

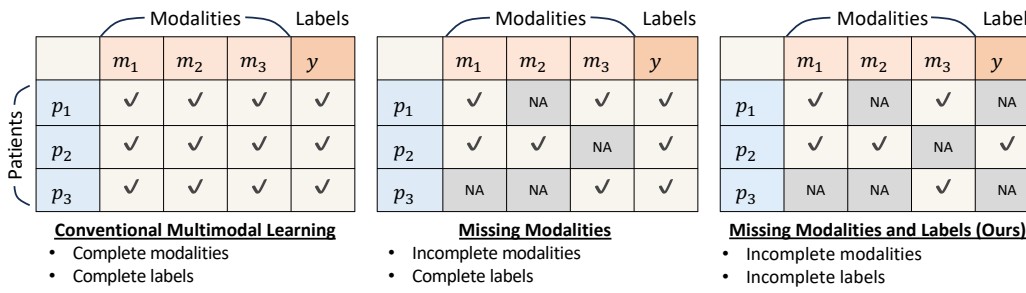

Figure 1: Existing multimodal learning works either presume the availability of all modalities for each patient or only deal with missing modalities. However, in reality, patient data usually comes with missing modalities and labels due to variations in clinical practice, linkage issues, legal and privacy concerns, or data corruption.

Shang et al., 2017) approaches leverage generative models to reconstruct the absent modalities using available ones, while direct prediction (Tsai et al., 2019; Kim et al., 2021) approaches use specialized network designs to perform the downstream task in the presence of missing modalities.

However, besides modalities, patient data can also miss label information. Alzheimer's patients, for example, might not have the disease progression label due to inconsistent follow-ups or patient dropout (Schott & Bartlett, 2012). We term this new problem as **multimodal learning with missing modalities and labels**. Fig. 1 provides a comparison among these three settings. To solve this problem, we identify the following challenges:

- **Missing Modalities and Labels.** Modality imputation approaches often rely on strong assumptions about data distribution and fail to fully use inter-patient relationships. On the other hand, direct prediction approaches either oversimplify the patient-modality relationship or suffer from scalability issues when adapting to varied modality absence patterns (You et al., 2020). Furthermore, these methods only deal with the missing modality issue and usually require supervision signals from labels. As a result, existing methods cannot leverage the information from the patients without labels.

- **Modality Collapse.** Multimodal models tend to over-rely on a subset of modalities during model training and ignore potentially valuable information from other modalities (Javaloy et al., 2022). As a result, existing models often do not generalize well to samples missing those key modalities.

To overcome these challenges, we introduce MUSE, a mutual-consistent graph contrastive learning framework. In essence, MUSE incorporates the following components:

- **Bipartite Patient-Modality Graph for Relationship Modeling.** To address the challenge of missing modalities, we construct a bipartite graph of two distinct node sets: patients and modalities. The modality features serve as links between patient and modality nodes. This dynamic graph structure supports patients with diverse modalities and facilitates efficient information transfer between modalities and patients.

- **Mutual-Consistent Contrastive Loss for Representation Learning.** Addressing the modality collapse problem and the challenge of missing labels, we introduce a mutual-consistent contrastive loss. The unsupervised contrastive objective encourages similar representations for the same patient with different modalities, thereby learning modality-agnostic features. Meanwhile, the supervised contrastive objective further emphasizes the similarities among patients with the same label, thus learning label-decisive features. Notably, the unsupervised contrastive objective is trained with edge dropout augmentation and can seamlessly extend the training scope to patients with missing labels.

To assess the effectiveness of MUSE, we conduct experiments on three publicly available patient datasets: MIMIC-IV (Johnson et al., 2023), eICU (Pollard et al., 2018), and ADNI (Jack et al., 2008). Our findings show that MUSE outperforms all baselines when trained solely on patients with labels. Furthermore, MUSE+ enhances the absolute AUC-ROC score improvement by ∼4% by expanding the training to include patients without labels. We also conduct detailed analyses and ablation studies to investigate the factors contributing to the performance gain achieved by MUSE.

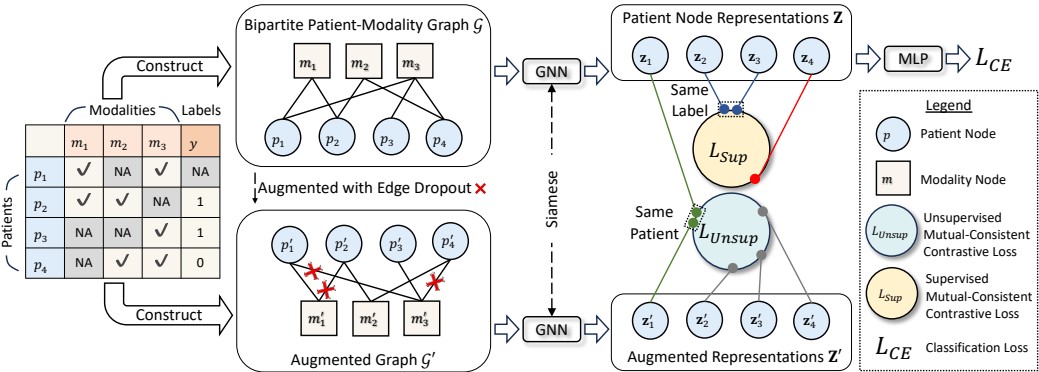

Figure 2: MUSE models the patient-modality relationship as a bipartite graph $\mathcal{G}$, where patients and modalities constitute two separate node sets, and modality features act as edges. An augmented version $\mathcal{G}'$ is then derived via edge dropout, signifying the same patients but with variations in modality inputs. The two graphs are encoded with a Siamese GNN to get the patient node representations. The unsupervised contrastive objective aims to maintain consistent patient representations across varied modalities, thereby learning modality-agnostic features. Meanwhile, the supervised contrastive objective promotes similarities between patients with identical labels, thereby learning label-decisive features. Note that during inference time, the augmentation branch is omitted.

## 2 PRELIMINARIES

Multimodal patient representation learning jointly models the interaction among multimodal observations for a patient, and learns comprehensive representations for downstream clinical predictive tasks. Formally, we use $\mathbf{X}^{(p)} = (\mathbf{x}_1^{(p)}, \mathbf{x}_2^{(p)}, \ldots, \mathbf{x}_M^{(p)})$ to denote the multimodal observation data (e.g., medical images, clinical reports, and biosignals) for the patient $p$, where $M$ is the number of modalities. And we use $\mathbf{y}^{(p)}$ to denote the label for the predictive tasks (e.g., hospital readmission, patient mortality, and disease diagnosis).

Conventional multimodal learning typically trains a multimodal model $f_\Theta(\cdot)$ with parameter $\Theta$ by minimizing a loss function $\mathcal{L}(\cdot)$ on a training set of $N$ patients $\mathcal{D}_{tr} = \{(\mathbf{x}_1^{(p)}, \ldots, \mathbf{x}_M^{(p)}, \mathbf{y}^{(p)})\}_{p=1}^N$, as in Eq. 1,

$$\arg\min_\Theta \mathbb{E}_{\mathcal{D}_{tr}}[\mathcal{L}(f_\Theta(\mathbf{x}_1^{(p)}, \ldots, \mathbf{x}_M^{(p)}), \mathbf{y}^{(p)})], \tag{1}$$

with the assumption that a complete set of modalities and labels is accessible to all patients. However, in real-world settings, patients can miss both modalities and labels due to various reasons. Formally, we represent the missingness in modalities as $\mathbf{M} \in \{0,1\}^{N \times M}$, where $\mathbf{A}[p, m] = 1$ if the patient $p$ has the modality $m$. And we use $\mathbf{L} \in \{0,1\}^N$ to denote the missingness in labels, where $\mathbf{L}[p] = 1$ if the label is available for the patient $p$. This paper then investigates: **How can we effectively learn representations for patients with missing modalities and labels?**

## 3 THE MUSE APPROACH

MUSE is driven by the following insights to learn multimodal representations for patients with missing data:

**Insight 1:** To handle patients with different modalities, the model must be flexible enough to accommodate different missing patterns and effective enough to model the patient-modal relationship. Inspired by recent works that utilize graphs to model the cross-data relationships (You et al., 2020; Chen & Zhang, 2020; Zhang et al., 2022), we represent the multimodal patient dataset as an undirected bipartite patient-modality graph $\mathcal{G}$. This graph represents patients and modalities as two types of nodes. An edge is drawn between a patient and a modality node if the patient possesses data for that modality. The feature of the modality functions as the edge's attribute. The flexibility of graph structure allows us to effectively and efficiently model the patient-modal relationship.

**Insight 2:** Intuitively, to address the modality collapse issue, the model needs to focus on features that are general across modalities and decisive for labeling. In this way, the model can still infer the label from modality-agnostic features learned from existing modalities, even if certain modalities are missing. Based on this intuition, we design a mutual-consistent contrastive loss. Specifically, we first obtain an augmented version of the bipartite graph $\mathcal{G}$ via edge dropout, denoted as $\mathcal{G}'$. The unsupervised contrastive goal is to align the same patient nodes in $\mathcal{G}$ and $\mathcal{G}'$. Meanwhile, a supervised contrastive loss is applied to enhance similarities between patients sharing the same label. Collectively, these objectives encourage the model to learn modal-agnostic and label-decisive features, thereby enhancing its generalizability. Moreover, the unsupervised contrastive loss only requires self-supervision signals, broadening the training scope to patients with missing labels.

Fig. 2 illustrates the MUSE framework. We will next describe the two modules in detail and introduce the training and inference strategy.

## 3.1 MULTIMODAL DATA AS A BIPARTITE GRAPH

Multimodal learning with graphs has attracted much attention recently due to their flexibility in modeling the interactions among modalities and samples (Ektefaie et al., 2023). However, existing approaches either only construct a graph for each modality (Zhang et al., 2022), which ignores cross-modal relationship, or build a hypergraph whose complexity increase as the combination of modalities (Chen & Zhang, 2020), which suffers from the scalable issue. In contrast, MUSE utilizes a bipartite patient-modality graph to maintain a good balance between effectiveness and scalability.

**Bipartite Patient-Modality Graph Construction.** MUSE transforms the multimodal patient features $\{\mathbf{X}^{(p)}\}_{p=1}^N$ and modality missingness matrix $\mathbf{M}$ to an undirected bipartite patient-modality graph $\mathcal{G} = (\mathcal{V}, \mathcal{E})$. Here, the node set $\mathcal{V} = \mathcal{V}_P \cup \mathcal{V}_M$ consists of patient nodes $\mathcal{V}_P = \{u_1, \ldots, u_N\}$ and modality nodes $\mathcal{V}_M = \{v_1, \ldots, v_M\}$. The edge set $\mathcal{E}$ is determined using the modality missingness matrix $\mathbf{M}$ as the adjacency matrix. Specifically, an edge $e_{u_p v_m}$ is drawn if the patient $p$ has the data for modality $m$.

To initialize edge embeddings, we utilize the associated modality features. Specifically, for the edge $e_{u_p v_m}$, we initialize its embedding by encoding the raw data $\mathbf{x}_m^{(p)}$ of modality $m$ pertaining to patient $p$, as in Eq. 2,

$$\mathbf{e}_{u_p v_m}^{(0)} = \text{Encoder}_m(\mathbf{x}_m^{(p)}), \tag{2}$$

where $\text{Encoder}_m(\cdot)$ is the backbone feature encoder for modality $m$, and $\mathbf{e}_{u_p v_m}^{(0)} \in \mathbf{R}^d$ is the encoded modality embedding with dimension $d$. For node embeddings, we initialize modality nodes as one-hot encodings of dimension $M$, i.e., $\mathbf{h}_{v_m}^{(0)} = \text{OneHot}(m)$. Patient nodes, on the other hand, are initialized with constant one vectors with the same dimension, i.e., $\mathbf{h}_{u_p}^{(0)} = \mathbf{1}$.

**Message Passing over the Bipartite Graph.** Given the constructed bipartite graph, we utilize a multi-layer Graph Neural Network (GNN) to integrate information from both modalities and patients, and learn consolidated patient representations, as in Eq. 3,

$$\mathbf{Z} = \text{GNN}(\mathcal{G}), \tag{3}$$

where $\mathbf{Z} \in \mathbb{R}^{N \times d}$ is the learned patient representations. Essentially, GNN propagates messages in the graph, where each node aggregates information from its neighboring nodes. Due to the multihop aggregation capability, GNNs can model long-term relations among modalities and patients. Specifically, in the $l$-th propagation layer, node $\mathbf{h}_i$ aggregates information from its neighboring nodes $\mathcal{N}(i)$ via the connecting edges, as in Eq. 4,

$$\mathbf{h}_i^{(l)} = \text{Aggregate}^{(l)} \left( \mathbf{h}_i^{(l-1)}, \left\{ \text{Message}^{(l)}(\mathbf{h}_j^{(l-1)}, \mathbf{h}_i^{(l-1)}, \mathbf{e}_{ji}^{(l-1)} \mid \forall j \in \mathcal{N}(i)) \right\} \right), \tag{4}$$

where $\text{Message}^{(l)}$ extracts message information from the source node representation $\mathbf{h}_j^{(l-1)}$, the target node representation $\mathbf{h}_i^{(l-1)}$, and the edge representation $\mathbf{e}_{ji}^{(l-1)}$. Meanwhile, $\text{Aggregate}^{(l)}$ compiles the neighboring messages via mean, sum, max, or other pooling operators.

### 3.2 MUTUAL-CONSISTENT CONTRASTIVE LEARNING

Multimodal features can be broadly categorized into two dichotomies: label-decisive versus non-label-decisive, and modality-agnostic versus modality-dependent (Xue et al., 2023). However, prior research on absent modalities does not differentiate between these features (You et al., 2020). As a result, a predictive model might over-rely on modality-dependent features, and yield compromised performance when the corresponding modalities are absent. For a model to consistently perform well across varying modalities, it is crucial to separate modality-agnostic and modality-dependent label-decisive features and focus on the modality-agnostic ones.

Based on this intuition, we introduce a simple yet effective mutual-consistent contrastive loss. Specifically, we first obtain an augmented version of the bipartite graph $\mathcal{G}$, through random edge dropout, which is denoted as $\mathcal{G}'$. This essentially mirrors the original set of patients but showcases different modality missing patterns. Subsequently, we pass the augmented graph $\mathcal{G}'$ through the same GNN encoder, yielding $\mathbf{Z}' = \text{GNN}(\mathcal{G}')$, where $\mathbf{Z}' \in \mathbb{R}^{N \times d}$ is the augmented patient representations. Building on this, MUSE introduces three loss objectives.

**Mutual-Consistent Contrastive Loss.** Patient nodes from graphs $\mathcal{G}$ and $\mathcal{G}$ represent the same set of patients but with different modalities as inputs. Naturally, we want to encourage the similarities between nodes from the same patient in different graphs to extract modality-agnostic features. To do this, MUSE deploys an unsupervised contrastive loss, as in Eq. 5,

$$\mathcal{L}_{\text{Unsup}}(\mathbf{Z}, \mathbf{Z}') = -\sum_{p=1}^{N} \log \frac{e^{s(\mathbf{z}_p, \mathbf{z}'_p)/\tau}}{\sum_{q=1}^{N} e^{s(\mathbf{z}_p, \mathbf{z}'_q)/\tau}}, \tag{5}$$

where $\mathbf{z}_p$ and $\mathbf{z}'_p$ are the nodes from the same patient in graph $\mathcal{G}$ and $\mathcal{G}'$, respectively, and $s(\cdot, \cdot)$ is the cosine similarity function with $\tau$ as a temperature hyperparameter. Notably, this loss function does not require label information to supervise the training. Thus, it can seamlessly incorporate patients with absent labels and extend the training scope.

While extracting modality-agnostic features is crucial, it is not the sole consideration. Patient data often encompass abundant auxiliary information irrelevant to the downstream task, such as administrative details. Thus, it is imperative to prioritize label-decisive features as well. In light of this, MUSE encourages patient nodes with the same label to predict each other. The corresponding loss is denoted as $\mathcal{L}_{\text{Sup}}(\mathbf{Z}, \mathbf{Y}, \mathbf{L})$, which is shown in Eq. 6,

$$-\sum_{p=1}^{N} \sum_{q=1}^{N} \mathbb{1}(\mathbf{L}[p]\mathbf{L}[q] = 1)\mathbb{1}(\mathbf{y}_p = \mathbf{y}_q) \log \frac{e^{s(\mathbf{z}_p, \mathbf{z}_q)/\tau}}{\sum_{r=1}^{N} \mathbb{1}(\mathbf{L}[r] = 1)\mathbb{1}(\mathbf{y}_p \neq \mathbf{y}_r)e^{s(\mathbf{z}_p, \mathbf{z}_r)/\tau}}, \tag{6}$$

where $\mathbb{1}(\cdot)$ is the indicator function. Here, $\mathbb{1}(\mathbf{L}[p]\mathbf{L}[q] = 1)$ and $\mathbb{1}(\mathbf{L}[r] = 1)$ only select patients with label information. And $\mathbf{z}_p$ and $\mathbf{z}_q$ denote patients with matching labels, whereas $\mathbf{z}_p$ and $\mathbf{z}_r$ signify patients with distinct labels.

**Classification Loss.** To further optimize for downstream tasks, we incorporate a supervised cross-entropy loss. Specifically, we first obtain the predicted label through a non-linear network, as $\hat{\mathbf{Y}} = \text{MLP}(\mathbf{Z})$, where $\text{MLP}(\cdot)$ is a multi-layer perceptron. Subsequently, we optimize the supervised classification loss, as $\mathcal{L}_{\text{CE}}(\hat{\mathbf{Y}}, \mathbf{Y}) = \sum_{p=1}^{N} \mathbb{1}(\mathbf{L}[p] = 1)\text{CE}(\hat{\mathbf{y}}_p, \mathbf{y}_p)$, where $\mathbb{1}(\mathbf{L}[p] = 1)$ selects patients with labels, and $\text{CE}(\cdot, \cdot)$ is the cross-entropy loss between predicted label $\hat{\mathbf{y}}_p$ and true label $\mathbf{y}_p$ for patient $p$.

### 3.3 TRAINING AND INFERENCE

To train MUSE, for each batch, we transform the batch data to a bipartite graph. Then, we employ an edge dropout of 0.15 probability to derive the augmented graph. Next, we input both the original and the augmented graph through the Siamese GNN. We leverage in-batch negatives for the contrastive losses in Eq. 5 and Eq. 6. Notably, similarity calculations are performed only once between each pair and are stored for reuse. The trio of loss objectives are added together to obtain the final loss. To infer MUSE, we only feed the original graph through the GNN and obtain the predicted label via the MLP classifier. The pseudocode of MUSE is available in Appx. A. More implementation details can be found in Sec. 4.1 and Appx. C.

# 4 EXPERIMENTS

## 4.1 EXPERIMENTAL SETUP

**Baselines.** We compare MUSE to three categories of baselines. (1) The first category is **modality imputation** methods, including CM-AE (Ngiam et al., 2011) and SMIL (Ma et al., 2021). (2) The second category is **direct prediction** approaches, including MT (Ma et al., 2022). (3) The last category is **graph-based** methods, including GRAPE (You et al., 2020), HGMF (Chen & Zhang, 2020), and M3Care (Zhang et al., 2022). Note that these methods depend on a supervision signal from task labels, and consequently, they only utilize labeled patients during training. To ensure a fair comparison, we implement our method in two distinct settings. When trained solely with labeled patients, it is referred to as MUSE; and when trained with both labeled and unlabeled patients, it is designated as MUSE+. More details can be found in appendix B.

**Implementations.** MUSE uses the GraphSage (Hamilton et al., 2017) with edge attributes version in You et al. (2020) as the GNN. The edge dropout rate is set to 15%. We split the dataset into 70%, 10%, 20% training, validation, and test sets. We train all models for 100 epochs on the training set, and select the best model by monitoring the performance on the validation set. The final results are reported on the test set. For each metric, we report the average scores and standard deviation by performing bootstrapping (i.e., sampling with replacement) 1000 times. Additionally, we conduct independent two-sample t-tests to assess whether MUSE achieves a significant improvement over the baseline methods. More implementation details are provided in Appx. C. The code of MUSE is publicly available [1].

## 4.2 RESULTS ON ICU DATA.

**Datasets.** We evaluate MUSE on two real-world ICU datasets. MIMIC-IV (Johnson et al., 2023) covers 431K visits for 180K patients admitted to the ICU in the Beth Israel Deaconess Medical Center. We use demographics (age, gender, and ethnicity), diagnosis, procedure, medication, lab values, and clinical notes as input modalities. eICU (Pollard et al., 2018) covers 431K visits for 180K patients admitted to the ICU in the Beth Israel Deaconess Medical Center. We use demographics, diagnosis, procedure, medication, lab values, and vital signals as input modalities.

**Tasks & Metrics**. We focus on two common clinical predictive tasks. Readmission prediction aims to predict whether a patient will be readmitted within the next 15 days following current discharge. Mortality prediction predicts whether a patient will pass away upon discharge in the eICU setting, or within 90 days after discharge in the MIMIC-IV setting. Both tasks are binary classifications. To evaluate the performance of the models, we calculate the Area Under the Precision-Recall Curve (AUC-PR) and the Area Under the Receiver Operating Characteristic Curve (AUC-ROC) scores.

**Backbone Encoders.** For demographic (age, gender, ethnicity) and sequential medical coding data (diagnosis, procedure, and medication), we use the Transformer (Vaswani et al., 2017) architecture as the backbone encoder. For times series data (lab values and vital signals), we use the Recurrent Neural Network (RNN). For text data (clinical notes), we use the TinyBERT (Jiao et al., 2019) encoder. We add a project layer to map the modality embeddings to the same latent space.

More details on datasets, statistics, tasks, metrics, and backbone encoders can be found in Appx. D.

**Results.** Table 1 presents the mortality and readmission prediction results on the MIMIC-IV and eICU datasets. Firstly, we observe that modality imputation methods CM-AE (Ngiam et al., 2011) and SMIL (Ma et al., 2021) perform the worst. This is reasonable as they rely on strong assumptions to learn the mapping from a lower-dimensional latent space to the higher-dimensional original input space. Direct prediction-based method, MT (Ma et al., 2022), offers slightly improved results by utilizing a specialized Transformer (Vaswani et al., 2017) architecture to model relationships among various modalities. However, it overlooks the valuable relation information among different patients. Next, graph-based approaches achieve the best performance among all baselines. These approaches model both inter- and intra-relationships between patients and modalities via GNNs. Within this category, HGMF (Chen & Zhang, 2020) performs the worst, likely due to its complex network design that might be challenging to optimize and prone to overfitting. Conversely, GRAPE (You et al.,

---

[1]`https://github.com/zzachw/MUSE`

Table 1: Results on the MIMIC-IV and eICU datasets. A dagger (†) indicates the standard deviation is greater than 0.02. An asterisk (*) indicates that MUSE achieves a significant improvement over the best baseline method, with a p-value smaller than 0.05. MUSE achieves the best performance across all baselines, and MUSE+ can further improve the performance by incorporating patients with missing labels into the unsupervised contrastive loss.

| Method | MIMIC-IV | | | | eICU | | | |
| | Mortality | | Readmission | | Mortality | | Readmission | |
| | AUC-ROC | AUC-PRC | AUC-ROC | AUC-PRC | AUC-ROC | AUC-PRC | AUC-ROC | AUC-PRC |
|---|---|---|---|---|---|---|---|---|
| CM-AE | 0.8530† | 0.4351† | 0.6817† | 0.4324† | 0.8624 | 0.3902 | 0.7462† | 0.4338† |
| SMIL | 0.8607 | 0.4438 | 0.6894† | 0.4368† | 0.8711 | 0.4066 | 0.7506 | 0.4447 |
| MT | 0.8739 | 0.4452 | 0.6901 | 0.4375 | 0.8882 | 0.4109 | 0.7635 | 0.4500 |
| GRAPE | 0.8837 | 0.4584† | 0.7085 | 0.4551 | 0.8903 | 0.4137 | 0.7663 | 0.4501 |
| HGMF | 0.8710 | 0.4433 | 0.7005† | 0.4421 | 0.8878 | 0.4104 | 0.7604 | 0.4496† |
| M3Care | 0.8896† | 0.4603† | 0.7067 | 0.4532 | 0.8964 | 0.4155 | 0.7598† | 0.4430 |
| MUSE | **0.9004*** | **0.4735*** | **0.7152†*** | **0.4670†*** | **0.9017*** | **0.4216*** | **0.7709*** | **0.4631*** |
| MUSE+ | **0.9201*** | **0.4883*** | **0.7351*** | **0.4985*** | **0.9332*** | **0.4387*** | **0.8003*** | **0.4844*** |

2020) and M3Care (Zhang et al., 2022) employ simpler graph designs and achieve better performance. Lastly, we observe that both MUSE and MUSE+ outperform all other baselines. Specifically, MUSE, trained on the same patient set as other baselines, achieves ∼2% absolute improvement in AUC-ROC score. Further amplifying this lead, MUSE+ elevates the improvement to ∼4% by incorporating additional patients with missing labels into the unsupervised contrastive loss.

## 4.3 RESULTS ON ALZHEIMER'S DISEASE DATA.

**Datasets.** The Alzheimer's Disease Neuroimaging Initiative (ADNI) (Jack et al., 2008) database provides longitudinal neuroimaging data, cognitive test scores, biomarker profiles, and genetic information for over 2K Alzheimer's disease, mild cognitive impairment, and normal. We use the processed data from the Alzheimer's Disease Prediction Of Longitudinal Evolution (TADPOLE) challenge (Marinescu et al., 2019). This challenge offers pre-processed features for 1737 patients spanning 12741 visits. Specifically, we utilize features from magnetic resonance imaging (MRI), positron emission tomography (PET), and diffusion tensor imaging (DTI).

**Tasks & Metrics**. The TADPOLE challenge involves classifying patient visits into three categories: normal control, mild cognitive impairment, and Alzheimer's disease. We adopt the official metrics: the balanced accuracy score and the one-vs-one macro AUC-ROC score.

**Backbone Encoders.** Given that the TADPOLE challenge supplies extracted features for each modality – like the volumes, thicknesses, and surface areas of region-of-interest (ROI) in MRI scans, or the mean, axial, and radial diffusivity in DTI ROIs – we implement a multi-layer perceptron equipped with ReLU activation and batch normalization as the backbone encoder.

More details on datasets, statistics, tasks, metrics, and backbone encoders can be found in Appx. E.

**Results.** Table 2 showcases the outcomes for predicting Alzheimer's disease progression using the ADNI dataset. We note that the standard deviations of the results are considerably larger than those in the ICU setting, with many having standard deviations > 0.02 (highlighted by the dagger †). This could likely be attributed to the smaller size of the dataset, resulting in a more volatile training process. However, the overall performance trend is similar to the ICU settings. The imputation-based methods, CM-AE (Ngiam et al., 2011) and SMIL (Ma et al., 2021), achieve the worst performance. Prediction-based method MT (Ma et al., 2022) achieves some improvement, while graph-based methods GRAPE (You et al., 2020) and M3Care (Zhang et al., 2022) achieve the best performance among all baselines. Particularly, we observe that HGMF (Chen & Zhang, 2020)'s performance further deteriorates on smaller dataset. Lastly, MUSE and MUSE+ give the best performance among all models, achieving ∼3% absolute improvements.

Table 2: Alzheimer's disease progression prediction results on the ADNI dataset. The dagger (†) and asterisk (*) have the same meanings as Tab. 1.

| Method | AUC-ROC | Accuracy |
|---|---|---|
| CM-AE | 0.8722† | 0.7305† |
| SMIL | 0.8761† | 0.7338† |
| MT | 0.8935 | 0.7604 |
| GRAPE | 0.9031† | 0.7820† |
| HGMF | 0.8845† | 0.7463† |
| M3Care | 0.9101 | 0.7822† |
| MUSE | **0.9158†*** | **0.7973†*** |
| MUSE+ | **0.9309*** | **0.8291*** |

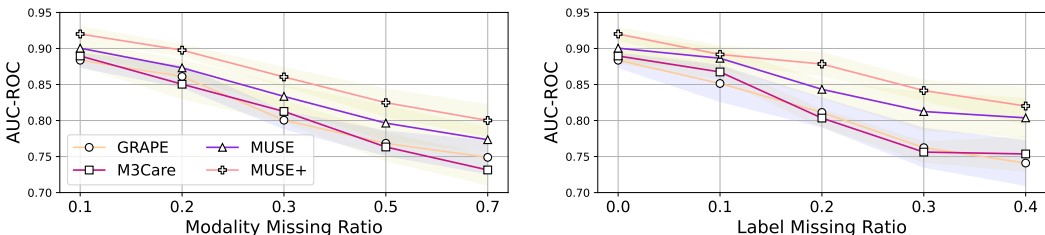

Figure 3: Experiment with different modality and label missing ratios.

## 4.4 ANALYSIS OF DIFFERENT DATA MISSING RATES.

Next, we analyze the impact of the data missing rate. Specifically, we randomly mask out modalities with probabilities {0.1, 0.2, 0.3, 0.5, 0.7}, and labels with probabilities {0, 0.1, 0.2, 0.3, 0.4}. We compare MUSE with the two top-performing baselines: GRAPE (You et al., 2020) and M3Care (Zhang et al., 2022) for the mortality prediction task on the MIMIC-IV dataset. The results are shown in Fig. 3. Results highlight that MUSE and MUSE+ consistently outperform the two baselines across all configurations. Moreover, we observe that as the missing ratio increases, the gap between MUSE+ and other baselines also increases. This demonstrates MUSE+'s robustness against missing data by incorporating complementary information from patients with missing labels.

## 4.5 ANALYSIS OF THE LEARNED REPRESENTATIONS.

Further, we study the quantitative evidence behind MUSE's success. Specifically, we measure the cosine similarity between representations from the same patient using the original modalities and those with 30% randomly masked modalities. Fig. 4 show the results calculated on the test set of the MIMIC-IV dataset for mortality prediction. We observe that the modality imputation-based methods (CM-AE (Ngiam et al., 2011) and SMIL (Ma et al., 2021)), as well as the direct prediction-based method (MT (Ma et al., 2022)) have much lower cosine similarities compared to the graph-based methods. This explains their performance drop as the patient representations vary a lot once certain modalities are missing. In contrast, MUSE and MUSE+ achieve the highest cosine similarity score among all baselines and also the best prediction performance.

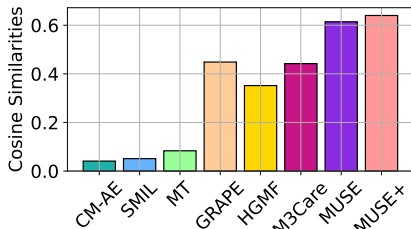

Figure 4: Cosine similarity of the representations from the same patients with different modalities.

These findings reaffirm our hypothesis regarding the efficacy of learning modality-agnostic representations.

## 4.6 ABLATION STUDY.

This section analyzes the performance lift brought by different components of MUSE. Specifically, A1 replaces edge dropout with feature dropout; A2 removes the mutual-consistent contrastive loss and only uses the cross-entropy loss; A3 removes the supervised contrastive loss; and A4 removes the unsupervised contrastive loss. The results can be found in Tab. 3. We observe that A1 suffers from the most significant performance drop, indicating that simple feature dropout cannot simulate the missing modality scenario. A2-4 remove different contrastive objectives, and all suffer from certain levels of performance drop. Specifically, A2 can be seen as special version of GRAPE (You et al., 2020) with additional input augmentation. It does not explicitly capture the modal-agnostic features and thus deleterious the most. A3 and A4 perform similarly, indicating that supervised and unsupervised mutual-consistent contrastive loss contributes differently to the overall objectives. This matches our intuition that the unsupervised contrastive loss forces the model to learn modal-agnostic features while the supervised version forces the model to learn label-decisive features.

Table 3: Ablation study on the influence of each individual component. The dagger (†) and asterisk (*) have the same meanings as Tab. 1.

| ID | Method | MIMIC-IV | | eICU | | ADNI |
|----|--------|----------|---|------|---|------|
| | | Mortality | Readmission | Mortality | Readmission | AD Progression |
| A1 | MUSE w/o Edge Dropout | 0.8830 | 0.6911† | 0.8655 | 0.7593 | 0.8811 |
| A2 | MUSE w/o Contrastive Loss | 0.8846 | 0.7055 | 0.8895 | 0.7611† | 0.9040 |
| A3 | MUSE w/o Supervised Contrastive Loss | 0.8934 | 0.7088† | 0.8916 | 0.7699 | 0.9086 |
| A4 | MUSE w/o Unsupervised Contrastive Loss | 0.8901 | 0.7103 | 0.8911 | 0.7706 | 0.9003 |
| - | MUSE | **0.9004*** | **0.7152†*** | **0.9017*** | **0.7709** | **0.9158†*** |
| - | MUSE+ | **0.9201*** | **0.7351*** | **0.9332*** | **0.8003*** | **0.9309*** |

## 4.7 RUNTIME ANALYSIS.

This section compares the prediction performance versus training time relationship for all methods. We report the per-epoch training time and AUC-ROC score for the MIMIC-IV mortality prediction task on a single NVIDIA A100 GPU. The results can be found in Fig. 5. MUSE achieves the best prediction performance with relatively small computation loads. MUSE+ further improves the prediction results with reasonable computation overhead (due to the incorporation of patients with missing label). This demonstrates that the design of MUSE keeps a good balance between effectiveness and efficiency. Furthermore, the simple definition of mutual-consistent contrastive loss allows re-using computations across various loss functions. These factors make MUSE a lightweight yet powerful framework.

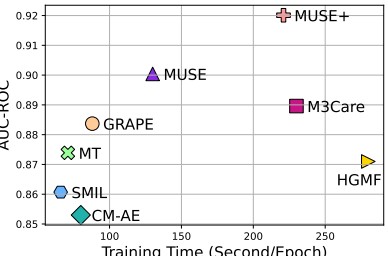

Figure 5: Prediction performance v.s. training time comparison on the MIMIC-IV mortality prediction task.

## 5 RELATED WORK

**Multimodal Learning with Missing Modalities.** Conventional multimodal approaches typically assume the availability of all modalities (Ramachandram & Taylor, 2017), which may not be satisfied in real-world scenarios. To address this, multimodal learning with missing modalities is proposed. CM-AE (Ngiam et al., 2011) leverages a cross-modality auto-encoder to recover the missed modalities based on existing ones. SMIL (Ma et al., 2021) further incorporates Bayesian meta-learning technique. However, these imputation-based methods tend to make strong assumptions about data distribution and may introduce additional noise. Furthermore, when imputing a missing modality for a certain patient, they fail to make full use of this modality from other patients. In contrast, MT (Ma et al., 2022) uses a late-fusion Transformer layer to model the interaction among modalities. MulT (Tsai et al., 2019) uses a pairwise Transformer layer to fuse the modality embeddings. And ViLT (Kim et al., 2021) and UMSE (Lee et al., 2023) directly enable the interaction among raw inputs from different modalities. These models can directly perform downstream tasks with the existence of missing modality due to specific network design. More recently, graph-based approaches have been proposed. GRAPE (You et al., 2020) utilizes a similar bipartite graph as MUSE, but it does not enforce the similarity of the same patient with different modalities. HGMF (Chen & Zhang, 2020) uses a heterogeneous graph to model different modality missing patterns. M3Care (Zhang et al., 2022) learns patient similarity for each modality. However, these methods either oversimplify the relationship, or utilize an overcomplex network that suffers from training and over-fitting issues. Furthermore, they require labels as supervisions and fail to utilize patients with missing labels in the training process.

## 6 CONCLUSION

Existing multimodal clinical predictive models either assume complete modality availability for each patient or only deal with missing modalities. In reality, both modalities and labels can be absent. To address this, we propose MUSE, which builds a patient-modality bipartite graph and learns to extract modality-agnostic and label-decisive feature via mutual-consistent graph contrastive learning. Evaluations of MUSE on three medical datasets reveal its superiority over baseline methods. Furthermore, we supplement our results with comprehensive qualitative analyses.

ACKNOWLEDGMENTS

This work was supported by NSF award SCH-2205289, SCH-2014438, and IIS-2034479. We thank the patients and their families who contributed to this research. This research was supported in part by the Intramural Research Program of the National Institute on Aging (NIA) and National Institute of Neurological Disorders and Stroke (NINDS), both part of the National Institutes of Health, within the Department of Health and Human Services project number ZIAAG000534. This work uses data obtained from the Alzheimer's Disease Neuroimaging Initiative (ADNI) database. ADNI is funded by the National Institute on Aging, the National Institute of Biomedical Imaging and Bioengineering, and through generous contributions from the following: AbbVie, Alzheimer's Association; Alzheimer's Drug Discovery Foundation; Araclon Biotech; BioClinica, Inc.; Biogen; Bristol-Myers Squibb Company; CereSpir, Inc.; Cogstate; Eisai Inc.; Elan Pharmaceuticals, Inc.; Eli Lilly and Company; EuroImmun; F. Hoffmann-La Roche Ltd and its affiliated company Genentech, Inc.; Fujirebio; GE Healthcare; IXICO Ltd.;Janssen Alzheimer Immunotherapy Research & Development, LLC.; Johnson & Johnson Pharmaceutical Research & Development LLC.; Lumosity; Lundbeck; Merck & Co., Inc.;Meso Scale Diagnostics, LLC.; NeuroRx Research; Neurotrack Technologies; Novartis Pharmaceuticals Corporation; Pfizer Inc.; Piramal Imaging; Servier; Takeda Pharmaceutical Company; and Transition Therapeutics. The Canadian Institutes of Health Research is providing funds to support ADNI clinical sites in Canada. Private sector contributions are facilitated by the Foundation for the National Institutes of Health (www.fnih.org). The grantee organization is the Northern California Institute for Research and Education, and the study is coordinated by the Alzheimer's Therapeutic Research Institute at the University of Southern California. ADNI data are disseminated by the Laboratory for Neuro Imaging at the University of Southern California. The investigators within the ADNI did not participate in analysis or writing of this manuscript.

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

**Contents of Appendix**

## A  Pseudocode of MUSE

---

**Algorithm 1:** Training and Inference for MUSE.

---

  1: *// Training*
**Require:** Training dataset $\mathcal{D}_{tr}$, modalities missingness matrix $\mathbf{M}$, label missingness matrix $\mathbf{L}$
  2: **for** each epoch **do**
  3:     **for** each batch **do**
  4:         Sample batch data from $\mathcal{D}_{tr}$
  5:         Transform batch data to bipartite graph $\mathcal{G}$
  6:         Initialize graph embeddings by Eq. 2
  7:         Obtain augmented graph $\mathcal{G}'$ via edge dropout
  8:         Feed both graphs through Siamese GNN by Eq. 3
  9:         Compute mutual-consistent contrastive loss by Eq. 5, 6
 10:         Obtain the predicted label via MLP and calculate the cross-entropy loss
 11:         Update model parameters with the weighted sum of all losses
 12:     **end for**
 13: **end for**
 14: *// Inference*
**Require:** Testing dataset $\mathcal{D}_{te}$, modalities missingness matrix $\mathbf{M}$
 15: **for** each batch **do**
 16:     Sample batch data from $\mathcal{D}_{te}$
 17:     Transform batch data to bipartite graph $\mathcal{G}$
 18:     Initialize graph embeddings by Eq. 2
 19:     Feed graph $\mathcal{G}$ through GNN by Eq. 3
 20:     Obtain the predicted label via MLP
 21: **end for**

---

## B  Additional Details on Baselines

We compare MUSE with the following baselines:

- **CM-AE (Ngiam et al., 2011)** leverages a cross-modality auto-encoder to recover the missed modalities. Specifically, we first sample a subset of patients with complete modalities. Then, we train CM-AE to reconstruct the masked-out modality. Next, we use the trained CM-AE model to impute missing modalities for all patients. Finally, we perform the downstream tasks on the imputed dataset.

- **SMIL (Ma et al., 2021)** further incorporates Bayesian meta-learning techniques to perturb the latent feature space so that embeddings with missing modalities can approximate the ones with complete modalities. Specifically, it approximates missing modality with a weighted sum of modality priors learned from the modality-complete dataset together with a feature regularizer.

- **MT (Ma et al., 2022)** utilizes a Transformer layer to fuse the embeddings from different modalities. Specifically, we use the late-interaction version where the Transformer fuses the learned modality embeddings from modality-specific encoders.

- **GRAPE (You et al., 2020)** utilizes a similar bipartite graph as MUSE to fuse the information across modalities and patients. We directly feed the fused patient representations from GNN to a downstream MLP for the predictive tasks.

- **HGMF (Chen & Zhang, 2020)** uses a heterogeneous hypernode graph to model the patient-modality relationship and to accommodate different modality missing patterns. Modalities from the same patients are modeled via an intra-node encoder. And the interaction among patients is modeled with the hypergraph.

- **M3Care (Zhang et al., 2022)** learns patient similarity for each modality, and then builds a per-modality similarity graph. The cross-patients interaction is modeled via a GNN. And the modalities embeddings for each patient are aggregated via a Transformer head.

## C   ADDITIONAL DETAILS ON IMPLEMENTATIONS

We train all models for 100 epochs on the training set, and select the best model by monitoring the performance on the validation set. The final results are reported on the test set. The specific metric monitored varies depending on the task: we used the PR-AUC score for mortality and readmission tasks and the one-vs-one macro AUC-ROC score for the Alzheimer's disease prediction task. MUSE uses the GraphSage (Hamilton et al., 2017) with edge attributes version in You et al. (2020) as the GNN. Specifically, in the $l$-th layer propagation layer, the node $\mathbf{h}_i$ aggregates information from its neighbours nodes via the connecting edges, as in Eq. 7,

$$\mathbf{h}_i^{(l)} = \mathbf{U}^{(l)} \text{Concat}\left(\mathbf{h}_i^{(l-1)}, \text{ Mean}\left(\left\{\text{ReLU}(\mathbf{W}^{(l)}\mathbf{h}_j^{(l-1)} + \mathbf{O}^{(l)}\mathbf{e}_{ji}^{(l-1)} \mid \forall j \in \mathcal{N}(i))\right\}\right)\right), \quad (7)$$

where $\mathbf{U}$, $\mathbf{W}$, $\mathbf{O}$ are learnable parameters. And the edge embeddings are updated as well, as in Eq. 8,

$$\mathbf{e}_{ji}^{(l)} = \mathbf{P}^{(l)} \text{Concat}\left(\mathbf{h}_j^{(l-1)}, \mathbf{h}_i^{(l-1)}\mathbf{e}_{ji}^{(l-1)}\right), \quad (8)$$

where $\mathbf{P}$ is a learnable parameter matrix. We set the number of GNN layers to 2. And we add a MLP classification head to each of the GNN layer. The cosine similarity temperature $\tau$ is set to 0.05. We would like to note that we did not extensively tune the hyperparameters for the experiments. Specifically, we only tune the learning rate for both MUSE and the baseline methods while keeping the other hyperparameters aligned to ensure a fair comparison. We implement MUSE using PyTorch Paszke et al. (2019) 1.11 and Python 3.8. The model is trained on a CentOS Linux 7 machine with 128 AMD EPYC 7513 32-Core Processors, 512 GB memory, and eight NVIDIA RTX A6000 GPUs. The code of MUSE will be released after acceptance.

## D   ADDITIONAL DETAILS ON EHR DATA

Table 4: ICU Dataset statistics.

| Item | MIMIC-IV | eICU |
|---|---|---|
| #Patients | 172113 | 120892 |
| #Admissions | 389761 | 158248 |
| Mortality Positive % | 0.09 | 0.04 |
| Mortality Missing % | 0.58 | 0.50 |
| Readmission Positive % | 0.25 | 0.15 |
| Readmission Missing % | 0.41 | 0.50 |
| Age | 57 | 62 |
| Gender | F: 0.52, M: 0.48 | F: 0.46, M: 0.54 |
| Race | African American: 0.17, Asian: 0.04, Caucasian: 0.66, Hispanic: 0.06, Native American: 0.00, Other: 0.06 | African American: 0.11, Asian: 0.02, Caucasian: 0.77, Hispanic: 0.04, Native American: 0.01, Other: 0.06 |
| Diagnosis Missing % | 0.00 | 0.08 |
| Procedure Missing % | 0.50 | 0.18 |
| Medication Missing % | 0.17 | 0.17 |
| Lab Missing % | 0.22 | 0.03 |
| Notes Missing % | 0.31 | - |
| Vital Signal Missing % | - | 0.11 |

### D.1   DATASETS

**MIMIC-IV (Johnson et al., 2023).** We extract the data from the patients, admissions, prescriptions, diagnoses_icd, procedures_icd, labevents, and discharge tables. Prescriptions, diagnoses, and procedures are recorded as medical codes. For labevents, following Sung et al. (2021), we gather ten groups of data related to Hematocrit, Platelet, WBC, Bilirubin, pH, Bicarbonate, Creatinine, Lactate, Potassium, and Sodium. A total of 111 lab items are included. For clinical notes, we use the text from the discharge summary section. We select our cohorts by filtering out visits of patients younger than 18 or older than 89 years old, and visits that last longer than 10 days. We exclude visits where the patient ultimately passed away, as the predictions are made upon discharge.

**eICU (Pollard et al., 2018).** We extract data from the patient, diagnosis, treatment, medication, lab, and apacheApsVar tables. Diagnosis, treatment, and medication are recorded as medical codes. We select our cohorts by filtering out visits of patients younger than 18 or older than 89 years old, visits that last longer than 10 days, and visits lasting shorter than 12 hours, as the predictions are made 12 hours after admission.

### D.2 TASKS

We focus on two common clinical predictive tasks: readmission prediction and mortality prediction.

For the MIMIC-IV dataset, the predictions are made at the time of discharge. Mortality prediction is defined as predicting whether a patient will pass away within 90 days after discharge. Readmission prediction aims to determine whether a patient will be readmitted within the next 15 days following discharge. We extract the mortality label from the date of death information. For patients without date of death information, we further check their next admission date. If their next admission date is after 90 days, we set the mortality label to false. Otherwise, we set the mortality label as missing. For the readmission label, we compare the current discharge date versus the next admission date. We set the readmission label to missing for each patient's last visit, since it is possible that the patient is transferred to a different hospital.

In the case of the eICU dataset, the predictions are made 12 hours after admission. Mortality prediction aims to predict whether a patient will pass away upon ICU discharge. Readmission prediction, on the other hand, aims to determine whether a patient will be readmitted to ICU within the next 15 days during the same hospital stay. Both mortality and readmission labels are extracted from the current hospital stay, and there are no missing labels. Thus, we simulate the missing label scenario by masking out 50% labels randomly.

### D.3 METRICS

We use the metric functions from the sklearn library. The AUC-ROC and AUC-PRC scores are calculated as follows:

```python
from sklearn import metrics

auc_roc = metrics.roc_auc_score(y_true, y_score)
precision, recall, thresholds = metrics.precision_recall_curve(
    ↪ y_true, y_score)
auc_prc = metrics.auc(recall, precision)
```

### D.4 BACKBONE ENCODERS

For demographic (age, gender, ethnicity) and sequential medical coding data (diagnosis, procedure, and medication), we use the Transformer (Vaswani et al., 2017) architecture as the backbone encoder. The number of layers is 2, the embedding dimension is 128, and the number of attention heads is 2. The event embedding look-up table is initialized with ClinicalBERT Alsentzer et al. (2019) embeddings of the event name and then projected down to 128 dimension with a linear layer. Patient demographics features (age, gender, and ethnicity) are separately embedded with another embedding look-up table. The medical and patient demographics are added together to form the overall sequence embedding. For times series data (lab values and vital signals), we use the Recurrent Neural Network (RNN). We set the number of layers to 2 and enable bidirectional propagation. For text data (clinical notes), we use the TinyBERT (Jiao et al., 2019) encoder. We set the maximum length to 256. We add a project layer to map the modality embeddings to the same latent space of dimension 128.

## E ADDITIONAL DETAILS ON AD DATA

### E.1 DATASETS

**ADNI (Jack et al., 2008).** We use the TADPOLE_D1_D2 table from the TADPOLE data challenge (Marinescu et al., 2019), which integrates data from the ADNIMERGE spreadsheet as well as additional MRI, PET (FDG, AV45 and AV1451), DTI and CSF biomarkers. Specifically, we use the MRI biomarker features from the UCSFFSL_02_01_16 and UCSFFSL51ALL_08_01_16 tables, the PET ROI biomarker features from the BAIPETNMRC_09_12_16, UCBERKELEYAV45_10_17_16 and UCBERKELEYAV1451_10_17_16 tables, and ROI summary measures (e.g. mean diffusivity MD, axial diffusivity AD) from the DTIROI_04_30_14 table.

Table 5: AD Dataset statistics.

| Item | ADNI |
|------|------|
| #Patients | 1572 |
| #Admissions | 5974 |
| Label Distribution | Normal Control: 0.03, Mild Cognitive Impairment: 0.46, Alzheimer's Disease: 0.22 |
| Label Missing % | 0.08 |
| Age | 74 |
| Gender | F: 0.45, M: 0.55 |
| Race | Indian/Alaskan: 0.00, Asian: 0.02, Black: 0.04, Hawaiian/Other PI: 0.00, More than one: 0.01, Unknown: 0.00, White: 0.93 |
| DTI Biomarkers Missing % | 0.87 |
| PET Biomarkers Missing % | 0.65 |
| MRI Biomarkers Missing % | 0.27 |

### E.2 TASKS

We focus on the Alzheimer's disease progression prediction task. The goal is to classify each patient visit into normal control, mild cognitive impairment, and Alzheimer's disease. The label information is extracted from the TADPOLE_D1_D2 table. There are no missing labels. Thus, we simulate the missing label scenario by randomly masking out 50% of the labels.

### E.3 METRICS

We adopt the official metrics: the balanced accuracy score and the one-vs-one macro AUC-ROC score. They are calculated as follows:

```python
from sklearn import metrics

b_acc = metrics.balanced_accuracy_score(y_true, y_pred)
auc_roc_macro_ovo = metrics.roc_auc_score(y_true, y_score, average
    ↪ ="macro", multi_class="ovo")
```

### E.4 BACKBONE ENCODERS

Given that the TADPOLE challenge supplies extracted features for each modality, we implement a two-layer perceptron with ReLU activation and batch normalization as the backbone encoder. The hidden dimension is set to 128.

## F ADDITIONAL EXPERIMENTS

### F.1 ADDITIONAL EVALUATION ON THE MISSING NOT AT RANDOM SETTING

In this paper, we follow existing works Zhang et al. (2022); You et al. (2020) and mainly focus on the missing completely at random setting. The modalities we consider are typically collected routinely for ICU patients, and ideally, they should be available for each patient. Missingness in our context is primarily attributed to administrative issues or time gaps.

Additionally, we simulate a scenario where label missingness is correlated with modality missingness and the true label for the eICU mortality prediction task. The results can be found in Tab. 6. Results show that MUSE can also adapt to this non-random missingness scenario as the correlation could be learned in a data-driven manner.

### F.2 ADDITIONAL ANALYSIS OF THE USE OF THE SUPERVISED CONTRASTIVE LOSS

We would like to note that the supervised contrastive loss and the classification loss are calculated on two different tensors. Specifically, we add a projection layer to the patient embedding before calculating the contrastive loss, while the classification loss is based on the original patient embedding. Our intuition behind this design is that the contrastive losses encourage the model to learn both

Table 6: Additional results on the eICU mortality prediction task under the missing not at random setting. We simulated this setting by assigning a higher label-missing rate to deceased patients with missing vital signals while keeping the overall label-missing rate still at 50%.

| Model | AUC-ROC |
|-------|---------|
| CM-AE | 0.8140 |
| SMIL | 0.8121 |
| MT | 0.8410 |
| GRAPE | 0.8698 |
| HGMF | 0.8511 |
| M3Care | 0.8700 |
| MUSE | **0.8753** |
| MUSE+ | **0.8911** |

Table 7: Additional ablation study on the supervised contrastive loss in terms of the AUC-ROC score.

| ID | Method | MIMIC-IV | | eICU | | ADNI |
|----|--------|----------|----|------|----|------|
| | | Mortality | Readmission | Mortality | Readmission | AD Progression |
| A3 | MUSE w/o Supervised Contrastive Loss | 0.8934 | 0.7088 | 0.8916 | 0.7699 | 0.9086 |
| A5 | A3 w/ higher Classification Loss | 0.8955 | 0.7002 | 0.8902 | 0.7713 | 0.9033 |
| - | MUSE | **0.9004** | **0.7152** | **0.9017** | **0.7709** | **0.9158** |
| - | MUSE+ | **0.9201** | **0.7351** | **0.9332** | **0.8003** | **0.9309** |

modality-agnostic and label-decisive features. In contrast, the classification loss leverages model-specific and label-decisive features to enhance prediction performance. To empirically support this, we provide additional ablation study on the supervised contrastive loss in terms of the AUC-ROC score in Tab. 7. This ablation study demonstrates the contribution of the supervised contrastive loss to the overall model performance.

### F.3 ADDITIONAL METRIC OF THE DISTANCE BETWEEN PATIENT REPRESENTATIONS

In our experiment analyzing the learned representations (Sec. 4.5), we calculate the average cosine similarity across all patients. Additionally, we include the average Euclidean distance between representations in Tab. 8.

Table 8: Euclidean distance of the representations from the same patients with different modalities.

| Model | Euclidean Distance |
|-------|--------------------|
| CM-AE | 0.7352 |
| SMIL | 0.6337 |
| MT | 0.5839 |
| GRAPE | 0.3001 |
| HGMF | 0.4426 |
| M3Care | 0.3286 |
| MUSE | **0.2437** |
| MUSE+ | **0.2232** |

## G LIMITATIONS

Our paper primarily addresses the challenge of handling missing modalities and labels. However, it is important to note that for a given task, there is no guarantee that multimodal models will consistently outperform unimodal methods. Investigating the conditions under which multimodal models

outperform unimodal ones is still an open research question and falls outside the scope of this work. Instead, our focus is on scenarios where multimodal models have the potential to outperform unimodal models, but their performance is limited by the absence of certain modalities and labels. For other applications, users should assess the necessity of employing multimodal learning before considering the application of our proposed method to address the issue of missing modalities and labels.

## H  NOTATIONS

| Notation | Meaning |
|---|---|
| $\mathbf{X}^{(p)}$ | multimodal observation data for patient $p$ |
| $(\mathbf{x}_1^{(p)}, \mathbf{x}_2^{(p)}, \dots, \mathbf{x}_M^{(p)})$ | $M$ multimodal observations for patient $p$ |
| $p$ | a specific patient |
| $m$ | a specific modality |
| $M$ | total number of modalities |
| $\mathbf{y}^{(p)}$ | label for patient $p$ |
| $f_\Theta(\cdot)$ | a multimodal model with parameter $\Theta$ |
| $\mathcal{L}(\cdot)$ | a general loss function |
| $\mathcal{D}_{tr} = \{(\mathbf{x}_1^{(p)}, \dots, \mathbf{x}_M^{(p)}, \mathbf{y}^{(p)})\}_{p=1}^N$ | a training set of $N$ patients |
| $\mathbf{M}$ | mortality missingness matrix |
| $\mathbf{A}[p, m]$ | whether patient $p$ has the modality $m$ |
| $\mathbf{L}$ | label missingness matrix |
| $\mathbf{L}[p]$ | whether the label is available for the patient $p$ |
| $\mathcal{G} = (\mathcal{V}, \mathcal{E})$ | a bipartite graph with node set $\mathcal{V}$ and edge set $\mathcal{E}$ |
| $\mathcal{V}_P = \{u_1, \dots, u_N\}$ | set of patient nodes |
| $\mathcal{V}_M = \{v_1, \dots, v_M\}$ | set of modality nodes |
| $e_{u_p v_m}$ | the edge between patient $p$ and modality $m$ |
| $\text{Encoder}_m(\cdot)$ | backbone feature encoder for modality $m$ |
| $\mathbf{e}_{u_p v_m}^{(0)}$ | initialized edge embedding |
| $\mathbf{h}_{v_m}^{(0)} = \text{OneHot}(m)$ | initialized modality node embedding |
| $\mathbf{h}_{u_p}^{(0)} = \mathbf{1}$ | initialized patient node embedding |
| $\text{GNN}(\cdot)$ | a graph neural network |
| $\mathbf{Z}$ | the learned patient representations |
| $\mathbf{h}_i^{(l)}, \mathbf{h}_i^{(l-1)}$ | embeddings for node $i$ at layer $l / l-1$ |
| $\mathbf{h}_j^{(l-1)}$ | embedding for node $j$ at leyer $l-1$ |
| $\mathbf{e}_{ji}^{(l-1)}$ | embedding for edge $e_{ji}$ between nodes $j$ and $i$ at layer $l-1$ |
| $\mathcal{N}(i)$ | neighboring nodes of $i$ |
| $\text{Message}^{(l)}$ | message extractor at layer $l$ |
| $\text{Aggregate}^{(l)}$ | message aggregator at layer $l$ |
| $\mathcal{G}'$ | an augmented version of graph $\mathcal{G}$ through edge dropout |
| $\mathbf{Z}'$ | the augmented patient representations |
| $\mathcal{L}_{\text{Unsup}}(\mathbf{Z}, \mathbf{Z}')$ | unsupervised mutual-consistent contrastive loss |
| $s(\cdot, \cdot)$ | cosine similarity function |
| $\tau$ | temperature hyperparameter |
| $\mathcal{L}_{\text{Sup}}(\mathbf{Z}, \mathbf{Y}, \mathbf{L})$ | supervised mutual-consistent contrastive loss |
| $\mathbb{1}(\cdot)$ | indicator function |
| $\mathbb{1}(\mathbf{L}[p]\mathbf{L}[q] = 1)$ | select patient $p$ and $q$ with labels |
| $\mathbb{1}(\mathbf{L}[r] = 1)$ | select patient $r$ with label |
| $\hat{\mathbf{Y}}$ | predicted labels |
| $\text{MLP}(\cdot)$ | a multi-layer perceptron |
| $\mathcal{L}_{\text{CE}}(\hat{\mathbf{Y}}, \mathbf{Y})$ | the classification loss |
| $\text{CE}(\cdot, \cdot)$ | the cross-entropy loss |

