# OpenReview forum: "Multimodal Patient Representation Learning with Missing Modalities and Labels"
_ICLR.cc/2024/Conference — ICLR 2024 poster_

### Official Review · Reviewer_D1yA · 2023-10-31

**Soundness:** 4 excellent
**Presentation:** 3 good
**Contribution:** 3 good
**Rating:** 6
**Confidence:** 4

**Summary:**

In this paper, the authors propose a novel framework, namely MUSE, to handle the missing modalities and labels while learning the multimodal representation of patients. The authors extend the conventional missing modalities problem to the missing labels issue and tackle both challenges concurrently using GNN-based contrastive learning. The authors conduct extensive experiments and analyses to demonstrate the superiority of the proposed MUSE.

**Strengths:**

1. The missing label problem widely exists in medical representation learning. However, previous works in this area do not handle it appropriately. This work provides excellent problem formulation to describe this research gap and proposes valuable solutions that can be inspirable to subsequent research.
2. The paper is well-organized. The research problem is described comprehensively. The necessity of emphasizing modality-agnostic and label-decisive features is clearly illustrated, solidifying the motivation of adopting supervised and unsupervised learning accordingly.
3. Modeling the modality missing and label missing problem is ingenious. Through contrastive learning, the proposed MUSE appropriately leverages alignments corresponding to these two problems, i.e., alignment between original and augmented graphs and between patients sharing different labels.
4. Experiments are extensive. Sections 4.4 – 4.7 significantly help readers to understand better how MUSE works, as well as advantages that cannot be simply reflected in major experiments in Sections 4.1 and 4.2.

**Weaknesses:**

1. More baselines should have been added for a more comprehensive comparison, e.g., MMIN (Zhao et al., Missing modality imagination network for emotion recognition with uncertain missing modalities.) and Robust Multimodal Transformer (Ma et al., Are multimodal transformers robust to missing modality?).
2. Other clinical tasks for ICU data can also be explored, such as ARF prediction and shock prediction (Tang et al.,  Democratizing EHR analyses with FIDDLE: a flexible data-driven preprocessing pipeline for structured clinical data.).

**Questions:**

Besides the concerns raised above, I have the following questions regarding this work:
1. If we regard labels as a subset of features, the whole architecture can be perceived as a pretraining framework combining two contrastive learning loss terms, along with a classification loss serving downstream tasks. I wonder if the authors have conducted experiments to pre-train MUSE with two contrastive learning terms and then fine-tune it using the classification loss term. The two-stage pattern might help the model to take a further step towards better performance by separating pretraining and fine-tuning.
2. For consistency, it is better to replace the “modality-general” in the abstract with “modality-agnostic.”
3. For readability, it might be better to explicitly introduce how the trio of losses are combined using an individual mathematical formula rather than verbally describing it in Section 3.3.

---

> ### Author Response · Authors · 2023-11-22
>
> We would like to thank the reviewer for the positive feedback. We have made the necessary updates to our manuscript. Below we would like to take this opportunity to respond to your questions.
>
> ---
>
> **Q1. Additional experiments with new datasets and baseline.**
>
> **A1.** In response to the reviewer's suggestion, we have conducted additional experiments on new datasets and baselines.
>
> > New datasets.
>
> We have conducted additional experiments on datasets from the MULTIBENCH [1] benchmark encompassing various areas such as Robotics, Finance, HCI, and Multimedia. Due to the time constraint, we selected one dataset from each area and evaluated our proposed MUSE and the two best-performing baselines (GRAPE and M3Care). To simulate the missing modalities and labels scenario, we randomly masked out labels with a probability of 0.5 and modalities with a probability of 0.1. In summary, our results indicate that MUSE performs exceptionally well on a variety of tasks and domains, demonstrating its generalizability.
>
> *Table: Additional experiment on the MULTIBENCH benchmark. MUSE achieves the best performance on 3/4 tasks, and MUSE+ further outperforms all baselines by incorporating samples with missing labels into the training process.*
>
> | Dataset | MUJOCO PUSH (Robotics) | STOCKS-F&B (Finance) | ENRICO (HCI) | MM-IMDB (Multimedia)  |
> |----------|----------|----------|--------|------------|
> | **Metrics** | **MSE $\downarrow$**  | **MSE $\downarrow$**    | **Acc $\uparrow$**  | **Micro F1 $\uparrow$**  |
> | GRAPE  | 8.0035  | 2.1165  | 0.3698 | 0.3851   |
> | M3Care  | 8.8192  | **2.0734**  | 0.3815 | 0.3882   |
> | MUSE (Ours)  | **7.5314**  | 2.0852  | **0.3887** | **0.3915**   |
> | MUSE+ (Ours)  | **5.6311**  | **2.0631**  | **0.4152** | **0.4232**   |
>
>
> > New baselines.
>
> We have extended our comparison by including three additional baselines: MedFuse [2], MMIN [3], and MMCL [4]. Notably, MMCL also incorporates samples with missing labels during training, making it a fair comparison with our enhanced method, MUSE+. The table below summarizes the performance comparison, with MUSE and MUSE+ consistently outperforming all baselines.
>
> *Table: Comparison between MUSE and three additional baselines in terms of AUC-ROC scores. MUSE and MUSE+ still outperform all baselines.*
>
> |      | MIMIC-IV Mortality   $\uparrow$   | MIMIC-IV Readmission  $\uparrow$ | eICU Mortality   $\uparrow$   | eICU  Readmission  $\uparrow$ | ADNI  $\uparrow$  |
> |------------|--------------------|--------------|-------------------|--------------|--------|
> | MedFuse  | 0.8733       | 0.6876    | 0.8813      | 0.7514    | 0.8916 |
> | MMIN    | 0.8712       | 0.6900     | 0.8756      | 0.7508    | 0.8844 |
> | MUSE  (Ours)  | **0.9004**       | **0.7152**    | **0.9017**      | **0.7709**    | **0.9158** |
> | MMCL    | 0.9134       | 0.7102    | 0.9192      | 0.7855    | 0.9218 |
> | MUSE+  (Ours)  | **0.9201**       | **0.7351**    | **0.9332**      | **0.8003**    | **0.9309** |
>
> ---
>
> **Q2. What if we pre-train MUSE with two contrastive learning terms and then fine-tune it using the classification loss term? What if we regard labels as a subset of features?**
>
> **A2.** We appreciate the thoughtful suggestion from the reviewer. Initially, we employed a two-stage training approach and subsequently merged them into an end-to-end training pipeline to improve efficiency. Empirically, we observed that both training schemes yield comparable performance, but the end-to-end approach significantly reduces training time. Regarding the idea of treating labels as a subset of features, we find it intriguing and believe it presents an interesting avenue for future research. We will include this idea in the future work section of the updated manuscript.
>
> ---
>
> Thank you once again for taking the time during the rebuttal phase. We hope that these updates address your questions.

---

### Official Review · Reviewer_eaku · 2023-10-31

**Soundness:** 3 good
**Presentation:** 3 good
**Contribution:** 3 good
**Rating:** 8
**Confidence:** 4

**Summary:**

The paper presents MUSE, a mutual-consistent graph contrastive learning method, which is designed to account for multimodal scenarios where both the label and different modalities are missing. This method is used in the context of clinical decision support and is validated on three different medical datasets (MIMIC-IV, eICU, and ADNI), against existing imputation and direct prediction approaches. Furthermore, the authors perform ablation experiments to better understand the value of each component in the model, examine the similarity of representations across methods, and discuss the run-time efficiency of methods. Overall, MUSE presents performance increases in the presented tasks without exceeding existing methods’ compute times.

**Strengths:**

+ The paper presents a strong argument for needing methods that can account for both missing modalities and missing labels. MUSE, based on the cited literature, is an original idea that has clear implications for the medical community and could be potentially extended to multimodal learning in other scenarios.


+ MUSE is validated on three datasets, which is important in demonstrating its generalizability. Furthermore, the additional experiments presented in the paper (missing data, ablation, similarity, and run-time analyses), add interesting insights into the different methods and are valuable in understanding what types of models work well in these multimodal medical settings.


+ The model architecture is straightforward and well-described. Representing patient-modality relationships in a graph and using edge dropout as a way to simulate missing modalities is creative yet easy to conceptualize. Incorporating both supervised and unsupervised contrastive learning is a great way to correct the modality collapse problem, as contrastive learning has been shown to be very effective in multimodal scenarios.


+ The authors describe the method in great detail and include the exact parameter values they used, which is a great community contribution as this method can be easily reproduced.

**Weaknesses:**

+ A weakness of the paper is that some of the architecture, experimental design, and parameter choices are not explained and not justified. For example, there are many ways to compute similarity (such as the normalized dot product used in the well-established InfoNCE loss), so a discussion on the choice of cosine as the similarity function is needed.

+ Similarly, while it is helpful for reproducibility to list the hyperparameters used in the model, this alone is not enough as their choice is not justified. Searching through the main paper as well as the appendix, there is no mention of hyperparameter tuning. Hyperparameter tuning is important both for the baselines and for the proposed MUSE to establish the fairness of comparison, as the wrong choice of parameter could lead to a baseline performing worse than it could. This is especially true as the baseline architectures are vastly different (ranging from auto-encoders to Transformers)

+ Other unexplained choices include using 15% as the edge dropout rate, the use of Siamese GNN over other GNN methods, doing the missing ratio experiment only on the mortality prediction task for MIMIC, and choosing 30% randomly masked modalities in the analysis of the learned representations.


+ Even if the MIMIC mortality prediction task is the most interesting, it would be great to include the same experiment for all other datasets and tasks in the appendix. Overall, anytime there is a choice of number, task, or a specific function, there should at least be a sentence explaining that choice, at a minimum backed by logic and intuition and ideally backed by experiments, to justify the omission of alternative choices.

+ The paper mentions the use of bootstrapping 1000 times, however, it is unclear if the reported performance is an average over the random weight initializations.


+ While the dataset statistics are mentioned in the Appendix, it is important to point out the class label imbalance present in the datasets in the main paper. This contextualizes the performance of the models; for example, MUSE+ performs with ~92% AUC-ROC on MIMIC mortality prediction, but only 9% of the data has a positive mortality prediction. Meaning, a model that does not learn truly learn the task, can simply guess the majority label and get 91% accuracy on the task. This may not be the case here, but this information is important to better understand performance.

Minor points:

+ The paper will benefit from a more comprehensive Related Works Section, not just on methods pertaining to missing modalities, but on other multimodal methods both supervised and unsupervised, and other contrastive loss functions. This would help the reader understand how the proposed formulas differ from existing formulas and why the author’s choices are better.

+ The authors do not discuss any limitations. The paper would benefit from a Limitations and a Discussion Section to address many of the aforementioned points.

**Questions:**

+ In the Missing Modalities and Labels paragraph, it is unclear what the text means by “fail to fully use inter-patient relationships?”

+ Could it be clarified what is meant by storing and reusing similarity calculations? Where is the similarity calculation reused, and why?

+ It is stated in the Training and Inference Section that for MUSE, the authors only feed the original graph through a GNN, but this contradicts the model diagram, which shows both augmented and original graphs going through a GNN. Can this be explained or clarified in the text?

+ The implementation details are unclear. “We train all models for 100 epochs on the training set, and select the best model by monitoring the performance on the validation set.” What is meant by “all models”? Does this include the baselines or only differently initialized MUSE models? Are the authors using the validation performance to determine all hyperparameters, and what specific metric is used? Is there early stopping for epochs, or are models trained for the full 100 epochs?


+ The paper cites the Transformer paper (Vaswani et al.) as the backbone encoder for several tabular modalities. The original Transformer paper cited is meant for text and not necessarily tabular data, so were there further adjustments made for this encoder?


+ In the Related Works Section, the authors mention MulT, ViLT, and UMSE, as other existing methods. Why were those methods not used as baselines?

---

> ### Author Response · Authors · 2023-11-22
>
> We would like to express our gratitude to the reviewer for their positive feedback. We have made the necessary updates to our manuscript. Below we would like to take this opportunity to respond to the reviewer's questions.
>
> ---
>
> **Q1. Design Choice & Hyperparameter Tuning.**
>
> **A1.** We would like to note that we did not extensively tune the hyperparameters for the experiments. Specifically, we only tune the learning rate for both MUSE and the baseline methods while keeping the other hyperparameters aligned. We leave extensive hyperparameters tuning for future works. As for design choice, we largely adhered to the settings presented in some of the classic papers, including the use of cosine similarity [1], a 15% edge drop rate [2], and the implementation of Siamese GraphSage [3]. It is important to note that we try to align the experimental settings between MUSE and other baselines as much as possible to ensure a fair comparison.
>
> ---
>
> **Q2. Additional experiments with new datasets and baseline.**
>
> **A2.** In response to the reviewer's suggestion, we have conducted additional experiments on new datasets and baselines.
>
> > New datasets.
>
> We have conducted additional experiments on datasets from the MULTIBENCH [1] benchmark encompassing various areas such as Robotics, Finance, HCI, and Multimedia. Due to the time constraint, we selected one dataset from each area and evaluated our proposed MUSE and the two best-performing baselines (GRAPE and M3Care). To simulate the missing modalities and labels scenario, we randomly masked out labels with a probability of 0.5 and modalities with a probability of 0.1. In summary, our results indicate that MUSE performs exceptionally well on a variety of tasks and domains, demonstrating its generalizability.
>
> *Table: Additional experiment on the MULTIBENCH benchmark. MUSE achieves the best performance on 3/4 tasks, and MUSE+ further outperforms all baselines by incorporating samples with missing labels into the training process.*
>
> | Dataset | MUJOCO PUSH (Robotics) | STOCKS-F&B (Finance) | ENRICO (HCI) | MM-IMDB (Multimedia)  |
> |----------|----------|----------|--------|------------|
> | **Metrics** | **MSE $\downarrow$**  | **MSE $\downarrow$**    | **Acc $\uparrow$**  | **Micro F1 $\uparrow$**  |
> | GRAPE  | 8.0035  | 2.1165  | 0.3698 | 0.3851   |
> | M3Care  | 8.8192  | **2.0734**  | 0.3815 | 0.3882   |
> | MUSE (Ours)  | **7.5314**  | 2.0852  | **0.3887** | **0.3915**   |
> | MUSE+ (Ours)  | **5.6311**  | **2.0631**  | **0.4152** | **0.4232**   |
>
>
> > New baselines.
>
> We have extended our comparison by including three additional baselines: MedFuse [2], MMIN [3], and MMCL [4]. Notably, MMCL also incorporates samples with missing labels during training, making it a fair comparison with our enhanced method, MUSE+. The table below summarizes the performance comparison, with MUSE and MUSE+ consistently outperforming all baselines.
>
> *Table: Comparison between MUSE and three additional baselines in terms of AUC-ROC scores. MUSE and MUSE+ still outperform all baselines.*
>
> |      | MIMIC-IV Mortality    $\uparrow$  | MIMIC-IV Readmission  $\uparrow$ | eICU Mortality  $\uparrow$    | eICU  Readmission  $\uparrow$ | ADNI  $\uparrow$  |
> |------------|--------------------|--------------|-------------------|--------------|--------|
> | MedFuse  | 0.8733       | 0.6876    | 0.8813      | 0.7514    | 0.8916 |
> | MMIN    | 0.8712       | 0.6900     | 0.8756      | 0.7508    | 0.8844 |
> | MUSE  (Ours)  | **0.9004**       | **0.7152**    | **0.9017**      | **0.7709**    | **0.9158** |
> | MMCL    | 0.9134       | 0.7102    | 0.9192      | 0.7855    | 0.9218 |
> | MUSE+  (Ours)  | **0.9201**       | **0.7351**    | **0.9332**      | **0.8003**    | **0.9309** |
>
> ---
>
> **Q3. Bootstrapping 1000 vs. Different Weight Initialization.**
>
> **A3.** We followed the evaluation protocol established in M3Care, which entails evaluating a trained model 1000 times by bootstrapping (i.e., sampling with replacement) from the test set.
>
> ---
>
> **Q4. Influence of Label Imbalance in Metrics.**
>
> **A4.** We understand the reviewer's concern regarding label imbalance. It is important to note the difference between accuracy and AUC-ROC scores. In datasets with only 1% positive labels, a classifier predicting all zeros would achieve a 99% accuracy score but only a 0.5 AUC-ROC score. Notably, AUC-ROC and PR-AUC scores are commonly used in mortality and readmission tasks on the MIMIC and eICU datasets [8, 9, 10].

---

> ### Author Response · Authors · 2023-11-22
>
> ---
>
> **Q5. In the Missing Modalities and Labels paragraph, it is unclear what the text means by “fail to fully use inter-patient relationships"?**
>
> **A5.** We appreciate the reviewer's feedback and recognize the need for clarification. Modality imputation approaches usually try to generate the missing modalities based on the available modalities from the same patient. However, we believe that the missing modalities could also be inferred from similar patients, while modality imputation approaches do not explicitly model the relationship between different patients.
>
> ---
>
> **Q6. Storing and Reusing Similarity Calculations.**
>
> **A6.** In a minibatch containing B samples, we calculate similarities between every pair of patients (resulting in B^2 similarity scores in total). These calculated similarities are reused both in the unsupervised and supervised contrastive loss functions. We will provide additional clarification in our revised manuscript.
>
> ---
>
> **Q7. It is stated in the Training and Inference Section that for MUSE, the authors only feed the original graph through a GNN, but this contradicts the model diagram, which shows both augmented and original graphs going through a GNN. Can this be explained or clarified in the text?**
>
> **A7.** We apologize for the confusion. The model diagram illustrates the training process, where both the augmented and original graphs pass through the GNN. During inference, the augmentation branch is omitted. We will revise our manuscript to make this distinction explicit.
>
> ---
>
> **Q8. The implementation details are unclear. “We train all models for 100 epochs on the training set, and select the best model by monitoring the performance on the validation set.” What is meant by “all models”? Does this include the baselines or only differently initialized MUSE models? Are the authors using the validation performance to determine all hyperparameters, and what specific metric is used? Is there early stopping for epochs, or are models trained for the full 100 epochs?**
>
> **A8.** To clarify, we trained both MUSE and the baseline methods for 100 epochs. We saved model weights at the end of each epoch and selected the best-performing model by monitoring its performance on the validation set. The specific metric monitored varies depending on the task: we used the PR-AUC score for mortality and readmission tasks and the one-vs-one macro AUC-ROC score for the Alzheimer's disease prediction task.
>
> ---
>
> **Q9. The paper cites the Transformer paper (Vaswani et al.) as the backbone encoder for several tabular modalities. The original Transformer paper cited is meant for text and not necessarily tabular data, so were there further adjustments made for this encoder?**
>
> **A9.** We use the Transformer encoder to encode the medical codes (e.g., diagnosis, procedures, and medications). However, they are sequence data instead of tabular. Each patient visit usually has a sequence of medical codes recorded at different timestamps. We will make this clear in the updated manuscript.
>
> ---
>
> Thank you for taking the time during the rebuttal phase. We hope that these updates address your questions.
>
> ---
>
> [1] SimCSE: Simple Contrastive Learning of Sentence Embeddings. EMNLP 2021.
>
> [2] BERT: Pre-training of Deep Bidirectional Transformers for Language Understanding. NAACL 2019.
>
> [3] Handling Missing Data with Graph Neural Networks. NeurIPS 2020.
>
> [4] MULTIBENCH: Multiscale Benchmarks for Multimodal Representation Learning. NeurIPS 2021.
>
> [5] MedFuse: Multi-modal fusion with clinical time-series data and chest X-ray images. MLHC 2022.
>
> [6] Missing Modality Imagination Network for Emotion Recognition with Uncertain Missing Modalities. ACL 2021.
>
> [7] Understanding Multimodal Contrastive Learning and Incorporating Unpaired Data. AISTATS 2023.
>
> [8] Multitask learning and benchmarking with clinical time series data. Nature Scientific Data 2019.
>
> [9] Learning the Graphical Structure of Electronic Health Records with Graph Convolutional Transformer. AAAI 2020.
>
> [10] Reproducibility in critical care: a mortality prediction case study. MLHC 2017.

---

> > ### Comment · Reviewer_eaku · 2023-11-22
> > **Thank you for response and clarification**
> >
> > Thank you for the detailed response to my comments.
> >
> > Could you please clarify why the different choices of evaluation metrics instead of calculating AUROC and AUPRC scores for all the tasks (Alzhiemers prediction versus mortality versus readmission)?
> >
> > Re. hyperparameter tuning, is it not a concern that a set of hyperparameters reported for a model for a certain dataset/task may not work as well when applied to a new dataset/task? Or, in this work, are the datasets/tasks are same across all baselines?

---

> > > ### Author Response · Authors · 2023-11-22
> > >
> > > Thank you for the comments! Please find our response below.
> > >
> > > ---
> > >
> > > **Q1. Choice of evaluation metrics for different tasks.**
> > >
> > > **A1.** The primary reason for using different evaluation metrics for these tasks is that the morality prediction and readmission tasks are binary classification problems, while the Alzheimer's disease prediction task involves multiclass classification (with three classes: normal control, mild cognitive impairment, and Alzheimer's disease). As a result, we are unable to calculate the PR-AUC for multiclass classification. Therefore, we have opted to use the official metrics specified in the TADPOLE data challenge (https://tadpole.grand-challenge.org/Performance_Metrics/), which include the balanced accuracy score and the one-vs-one macro AUC-ROC score.
> > >
> > > ---
> > >
> > > **Q2.Hyperparameter tuning.**
> > >
> > > **A2.** We understand your concern and would like to clarify our approach. We observed that the learning rate had the most significant impact on prediction performance in the pilot study. Thus, we conducted hyperparameter tuning to find the optimal learning rate for all baseline methods. We maintained consistency in other hyperparameters across the different methods to ensure a fair comparison.
> > >
> > > ---
> > >
> > > Please do not hesitate to let us know if you have any further questions. Thank you once again for taking the time during the rebuttal phase!

---

> > > > ### Comment · Reviewer_eaku · 2023-11-22
> > > >
> > > > Thank you for the clarification. Given the responses, I will raise my final score. I recommend that the authors include details about the evaluation metrics and hyperparameter tuning in the appendix.

---

### Official Review · Reviewer_3igL · 2023-10-31

**Soundness:** 3 good
**Presentation:** 4 excellent
**Contribution:** 4 excellent
**Rating:** 6
**Confidence:** 4

**Summary:**

This work creates a contrastive learning approach to learning patient representations across data modalities, and augments the learning procedures by creating bipartite patient-modality graphs with various dropout to represent missing data modalities. Through the contrastive objectives, it learns to keep similar representations aligned in the embedded space despite a variety of modality differences. Additionally the authors learn to represent similar and dissimilar representations in relation to downstream labels, in order to provide an ability to represent patients even without said labels. They then demonstrate their model effectiveness over a variety of tasks, compared against a number of baselines.

**Strengths:**

- The approach to learn representations that may allow for some variance in the features collected per person is an important finding for clinical applications

- There is novel application and design of the contrastive objectives, allowing for modality differences as well as using labels to find representations guided by the strongest risk factors

- There are a comprehensive set of baselines and the Muse+ model outperforms on all tasks

- The paper is extremely well written and clear to follow

**Weaknesses:**

Major

 - The use of the supervised contrastive loss and the classification loss seems redundant. This is essentially the cross-entropy loss of the representations followed by the cross-entropy loss of a non-linear transformation of the representation. I appreciate ablation study showing that the removal of the supervised contrastive loss results in worse performance. I see that the final loss is the unweighted sum of all the losses. Is it possible to achieve the same results by removing the supervised contrastive loss and adding a higher weight to the classification loss? If it is necessary, could you provide a justification for why we need to classification losses?

Minor
 - While ablation studies were provided that showed an array of dropout, did authors find a rate of missingness that ultimately resulted in the model ignoring certain features? (e.g. really rare lab examinations)
 - Typo in Appendix D.1 first paragraph. “For clinical nodes” should be “For clinical notes” I think.

**Questions:**

- Is it possible to achieve the same results by removing the supervised contrastive loss and adding a higher weight to the classification loss? If it is necessary, could you provide a justification for why we need to classification losses?

 - For the experiment analyzing the learned representation (Section 4.5) are you using the average cosine similarity across all patients? Could you also provide average Euclidean distance between representations since GRAPE minimizes MSE loss for continuous values?

---

> ### Author Response · Authors · 2023-11-22
>
> We sincerely appreciate the reviewer's valuable feedback and have made necessary updates to our manuscript. We would like to take this opportunity to address the questions and clarify some points regarding our manuscript.
>
> ---
>
> **Q1. The use of the supervised contrastive loss and the classification loss seems redundant.**
>
> **A1.** We understand the reviewer's concern about the perceived redundancy of the two supervised losses. We would like to clarify that the supervised contrastive loss and the classification loss are calculated on two different tensors. Specifically, we add a projection layer to the patient embedding before calculating the contrastive loss, while the classification loss is based on the original patient embedding. Our intuition behind this design is that the contrastive losses encourage the model to learn both modality-agnostic and label-decisive features. In contrast, the classification loss leverages model-specific and label-decisive features to enhance prediction performance. To empirically support this, we provide the following additional ablation study:
>
> *Table: Additional ablation study on the supervised contrastive loss in terms of the AUC-ROC score.*
>
>
> |            | MIMIC-IV Mortality  $\uparrow$ | MIMIC-IV Readmission  $\uparrow$ | eICU Mortality  $\uparrow$ | eICU Readmission  $\uparrow$ | ADNI  $\uparrow$  |
> |------------------------|-----------|----------|----------|----------|--------|
> | w/o Supervised Contrastive Loss | 0.8934  | 0.7088  | 0.8046  | 0.7699  | 0.7458 |
> | w/o Supervised Contrastive Loss, w/ higher Classification loss | 0.8955  | 0.7002  | 0.8124  | 0.7713  | 0.7495 |
> | MUSE          | **0.9004**  | **0.7152**  | **0.8264**  | **0.7709**  | **0.7533** |
> | MUSE+         | **0.9201**  | **0.7351**  | **0.8511**  | **0.8003**  | **0.7784** |
>
> This ablation study demonstrates the contribution of the supervised contrastive loss to the overall model performance.
>
> ---
>
> **Q2. Did authors find a rate of missingness that ultimately resulted in the model ignoring certain features?**
>
> **A2.** Yes. In fact, we observed that in the MIMIC-IV dataset, naive multimodal models (such as direct concatenation) frequently relied heavily on the clinical text while largely disregarding other modalities. We have referred to this issue as "Modality Collapse" in the introduction section of our manuscript. This observation was a critical motivator for developing the mutual-consistent graph contrastive learning framework. This framework is designed to facilitate the learning of features that are both modality-agnostic and decisive for accurate predictions.
>
> ---
>
> **Q3. Could you also provide the average Euclidean distance between representations since GRAPE minimizes MSE loss for continuous values?**
>
> **A3.** In our experiment analyzing the learned representations (Section 4.5), we calculate the average cosine similarity across all patients. We appreciate the suggestion to include the average Euclidean distance between representations. Here is the table with the requested information:
>
> *Table: Euclidean distance of the representations from the same patients with different modalities.*
>
> | Method      | Euclidean Distance  $\downarrow$ |
> |-------------------|--------------------|
> | CM-AE       | 0.7352       |
> | SMIL       | 0.6337       |
> | MT       | 0.5839       |
> | GRAPE       | 0.3001       |
> | HGMF       | 0.4426       |
> | M3Care      | 0.3268       |
> | MUSE (Ours)    | **0.2437**       |
> | MUSE+  (Ours)   | **0.2232**       |
>
> ---
>
> Thank you for taking the time during the rebuttal phase. We hope that these updates could address your questions.

---

### Official Review · Reviewer_se67 · 2023-11-06

**Soundness:** 3 good
**Presentation:** 3 good
**Contribution:** 2 fair
**Rating:** 6
**Confidence:** 3

**Summary:**

The authors tackle the problem of multimodal self-supervised representation learning using clinical data, where some modalities and some labels may be missing. They introduce a new contrastive earning method, which represents the dataset as a bipartite graph between patients and observed modalities. They initialize edge embeddings with observed data, and run a graph neural network over this graph, using a loss that combines supervised and unsupervised contrastive loss with the downstream classification loss. They evaluate their method on three clinical datasets, finding that they outperform the baselines.

**Strengths:**

- The proposed method is intuitive and easy to understand.
- The paper is generally well-written.
- The authors motivate the gain of jointly modelling missingness in modalities and labels both intuitively and empirically.

**Weaknesses:**

1. The datasets studied in the paper are fairly limited from a modality perspective in the classical sense (e.g. MIMIC-IV has time series and text, and eICU has only time series). Though the authors split the time-series into multiple modalities (e.g. labs, vitals, diagnoses), it is unclear what a modality has to encompass in order for the method to work well. In the extreme case, could each individual measurement have been its own modality?

2. The method proposed in the paper tackles a healthcare problem, but could definitely be applied to datasets beyond healthcare. The authors should consider benchmarking their method on datasets from [1], potentially by masking out labels and modalities.

3. It was not clear to me reading through the paper exactly how the method handles completely new patients at test-time. Are they added as new patient nodes and edges in the bipartite graph, and the GNN is run in inference with all of the pre-training patient nodes still in the graph?

4. The authors should add [2] as a baseline which does not make use of any labels during pretraining, which can then be finetuned on the labelled set. The authors should also add in baselines corresponding to pre-training an encoder on each modality separately (e.g. as in [3]), either with contrastive learning or with supervised learning, to show the gain of modelling modalities jointly in the pre-training stage.

5. In healthcare, missingness is often informative, e.g. a patient did not receive a chest X-ray because the physician did not suspect any pulmonary conditions, and such patients are thus less likely to have a pulmonary condition. The authors should consider probing this effect in the experiments. In particular, on the eICU dataset (where the authors currently add missingness to labels randomly), they could instead correlate label missingness with modality missingness and the true label.

[1] MULTIBENCH: Multiscale Benchmarks for Multimodal Representation Learning. NeurIPS 2021.

[2] Understanding Multimodal Contrastive Learning and Incorporating Unpaired Data. AISTATS 2023.

[3] MedFuse: Multi-modal fusion with clinical time-series data and chest X-ray images. MLHC 2022.

**Questions:**

Please address the weaknesses above. There is also a minor typo -- the second $\mathcal{G}$ should say $\mathcal{G}'$ in Section 3.2 paragraph "Mutual-Consistent Contrastive Loss" .

---

> ### Author Response · Authors · 2023-11-22
>
> We sincerely appreciate the valuable feedback provided by the reviewer. We have carefully considered the reviewer's comments and have made the necessary revisions to our manuscript. Below, we would like to address the reviewer's questions and provide additional information:
>
> ---
>
> **Q1. What should a modality encompass for multimodal models to work well?**
>
> **A1.** Generally speaking, we believe that this is still an open research question. I.e., when and why does multimodal outperform unimodal jointly? However, our paper primarily focuses on addressing the challenge of missing modalities and labels rather than delving into the theoretical aspects of modality selection. We designed our experiments based on real-world clinical practices, where various modalities play a role in clinical decision-making. We also empirically verified that multimodal models outperform unimodal methods in our setting during the pilot study. Understanding when and why multimodal models will work is out of the scope of this paper. Instead, we focus on the setting where multimodal models can outperform unimodal models, but the missing modalities and labels limit their improvements. We believe this question should not undermine the practical significance of the problem or the novelty of the proposed method.
>
> ---
>
>
> **Q2. Additional experiments on datasets beyond healthcare.**
>
> **A2.** In response to the reviewer's suggestion, we have conducted additional experiments on datasets from the MULTIBENCH [1] benchmark encompassing various areas such as Robotics, Finance, HCI, and Multimedia. Due to the time constraint, we selected one dataset from each area and evaluated our proposed MUSE and the two best-performing baselines (GRAPE and M3Care). To simulate the missing modalities and labels scenario, we randomly masked out labels with a probability of 0.5 and modalities with a probability of 0.1.
>
> *Table: Additional experiment on the MULTIBENCH benchmark. MUSE achieves the best performance on 3/4 tasks, and MUSE+ further outperforms all baselines by incorporating samples with missing labels into the training process.*
>
> | Dataset | MUJOCO PUSH (Robotics) | STOCKS-F&B (Finance) | ENRICO (HCI) | MM-IMDB (Multimedia)  |
> |----------|----------|----------|--------|------------|
> | **Metrics** | **MSE $\downarrow$**  | **MSE $\downarrow$**    | **Acc $\uparrow$**  | **Micro F1 $\uparrow$**  |
> | GRAPE  | 8.0035  | 2.1165  | 0.3698 | 0.3851   |
> | M3Care  | 8.8192  | **2.0734**  | 0.3815 | 0.3882   |
> | MUSE (Ours)  | **7.5314**  | 2.0852  | **0.3887** | **0.3915**   |
> | MUSE+ (Ours)  | **5.6311**  | **2.0631**  | **0.4152** | **0.4232**   |
>
> In summary, our results indicate that MUSE performs exceptionally well on a variety of tasks and domains, demonstrating its generalizability.
>
> ---
>
> **Q3. How does the method handle entirely new patients during testing?**
>
> **A3.** In both the training and inference phases, we create patient graphs independently within each batch. We will clarify this point in the revised manuscript.
>
> ---
>
> **Q4. Additional experiments on more baselines.**
>
> **A4.** As suggested by the reviewers, we have extended our comparison by including three additional baselines: MedFuse [2], MMIN [3], and MMCL [4]. Notably, MMCL also incorporates samples with missing labels during training, making it a fair comparison with our enhanced method, MUSE+. The table below summarizes the performance comparison, with MUSE and MUSE+ consistently outperforming all baselines.
>
> *Table: Comparison between MUSE and three additional baselines in terms of AUC-ROC scores. MUSE and MUSE+ still outperform all baselines.*
>
> |      | MIMIC-IV Mortality  $\uparrow$    | MIMIC-IV Readmission  $\uparrow$ | eICU Mortality  $\uparrow$    | eICU  Readmission  $\uparrow$ | ADNI  $\uparrow$  |
> |------------|--------------------|--------------|-------------------|--------------|--------|
> | MedFuse  | 0.8733       | 0.6876    | 0.8813      | 0.7514    | 0.8916 |
> | MMIN    | 0.8712       | 0.6900     | 0.8756      | 0.7508    | 0.8844 |
> | MUSE  (Ours)  | **0.9004**       | **0.7152**    | **0.9017**      | **0.7709**    | **0.9158** |
> | MMCL    | 0.9134       | 0.7102    | 0.9192      | 0.7855    | 0.9218 |
> | MUSE+  (Ours)  | **0.9201**       | **0.7351**    | **0.9332**      | **0.8003**    | **0.9309** |

---

> ### Author Response · Authors · 2023-11-22
>
> **Q5. Missingness is often informative. What if the labels are correlated with modality missingness?**
>
> **A5.** In this paper, we follow existing works (M3Care and GRAPE) and mainly focus on the missing completely at random setting. The modalities we consider are typically collected routinely for ICU patients, and ideally, they should be available for each patient. Missingness in our context is primarily attributed to administrative issues or time gaps.
>
> To address the reviewer's concern, we simulated a scenario where label missingness is correlated with modality missingness and the true label in the eICU mortality prediction task. In this setting, we found that our proposed method can adapt to the non-random missingness scenario. The table below presents the AUC-ROC scores under this condition. Results show that our proposed method can also adapt to this scenario as the correlation could be learned in a data-driven manner.
>
> *Table: Results on the eICU mortality prediction task under the missing not at random setting. We simulated this setting by assigning a higher label-missing rate to deceased patients with missing vital signals while keeping the overall label-missing rate still at 50%.*
>
> | Model | AUC-ROC  $\uparrow$ |
> |--------|--------|
> | CM-AE | 0.8140 |
> | SMIL  | 0.8121 |
> | MT  | 0.8410 |
> | GRAPE | 0.8698 |
> | HGMF  | 0.8511 |
> | M3Care | 0.8700 |
> | MUSE (Ours) | **0.8753** |
> | MUSE+ (Ours) | **0.8911** |
>
> ---
>
> We greatly appreciate your time and consideration during the rebuttal phase. We hope these updates could effectively address your questions and further enhance the quality and comprehensiveness of our manuscript.
>
> ---
>
>
> [1] MULTIBENCH: Multiscale Benchmarks for Multimodal Representation Learning. NeurIPS 2021.
>
> [2] MedFuse: Multi-modal fusion with clinical time-series data and chest X-ray images. MLHC 2022.
>
> [3] Missing Modality Imagination Network for Emotion Recognition with Uncertain Missing Modalities. ACL 2021.
>
> [4] Understanding Multimodal Contrastive Learning and Incorporating Unpaired Data. AISTATS 2023.

---

> > ### Comment · Reviewer_se67 · 2023-11-22
> >
> > Thank you for the response. The additional experiments have addressed most of my questions and concerns, and I have increased my score as a result. I would recommend the authors integrate these results into their revision.

---

### Author Response · Authors · 2023-11-22

We extend our sincere appreciation to all reviewers for dedicating their time and expertise during the rebuttal phase. Their constructive comments are invaluable to the enhancement of our manuscript.

To address the reviewers’ comments, we have included the following new results:
- 1. additional experiments on the MULTIBENCH benchmark
- 2. additional comparison with recent multimodal baselines
- 3. additional evaluation on the missing not at random setting
- 4. additional analysis of the use of the supervised contrastive loss
- 5. additional metric of the distance between patient representations

Furthermore, we have augmented our manuscript with detailed explanations of our model pipeline and experimental setup to enhance transparency and clarity.

We believe these revisions have significantly enriched our manuscript, aligning it more closely with the rigorous standards of this conference. We hope our revised manuscript could make a meaningful contribution to the field of machine learning for healthcare.

---

### Meta-Review · Area_Chair_NzwG · 2023-12-06

**Metareview:**

The paper proposes a new multimodal fusion method that handles missing modalities and labels. The reviewers find the method intuitive and well-written. They appreciated the flexibility of the representation learning and the use of contrastive objectives, as well as the extensive experiments. The reviewers had questions about the loss and the experiments, as well as the justification for the architecture, which the authors successfully addressed. Overall, following the discussion, this submission has emerged as a strong contribution to the area of multimodal learning, which is a topic of increased interest to the community.

**Justification For Why Not Higher Score:**

The paper is not on par with prior work that was selected for spotlights.

**Justification For Why Not Lower Score:**

Unanimous accept.

---

### Decision · Program_Chairs · 2024-01-16

Accept (poster)